# Challenges and Implications of the COVID-19 Pandemic on Mental Health: A Systematic Review

Abdullahi Rabiu Abubakar [1], Maryam Abba Tor [2], Joyce Ogidigo [3], Ibrahim Haruna Sani [4],
Adekunle Babajide Rowaiye [3], Mansur Aliyu Ramalan [1], Sani Yahaya Najib [5], Ahmed Danbala [6],
Fatima Adamu [7], Adnan Abdullah [8], Mohammed Irfan [9], Santosh Kumar [10], Ayukafangha Etando [11],
Sayeeda Rahman [12], Susmita Sinha [13] and Mainul Haque [14,*]

1   Department of Pharmacology and Therapeutics, Faculty of Pharmaceutical Sciences, Bayero University, Kano-700233, Kano PMB 3452, Nigeria; unisza7@gmail.com (A.R.A.); mmramalan@gmail.com (M.A.R.)
2   Department of Health and Biosciences, University of East London, University Way, London E16 2RD, UK; maryamabbator@yahoo.com
3   National Biotechnology Development Agency, Abuja 09004, Nigeria; jogidigo@yahoo.com (J.O.); adekunlerowaiye@gmail.com (A.B.R.)
4   Department of Clinical Pharmacology and Therapeutics, College of Health Sciences, Yusuf Maitama Sule University, Kofar Kansakali-700282, Kano PMB 3220, Nigeria; harunaibrahim81@yahoo.com
5   Department of Pharmaceutical and Medicinal Chemistry, Bayero University, Kano 700233, Nigeria; najibsani62@gmail.com
6   Department of Pharmacology and Toxicology, Faculty of Pharmaceutical Sciences, Kaduna State University, Kaduna 800283, Nigeria; ahmed.danbala@kasu.edu.ng
7   Department of Community Medicine, Aminu Kano Teaching Hospital, Kano 700233, Nigeria; fatimahrufaida@gmail.com
8   Unit of Occupational Medicine, Faculty of Medicine and Defence Health, Universiti Pertahanan Nasional Malaysia (National Defence University of Malaysia), Kuala Lumpur 57000, Malaysia; dradnanabdul-lah@upnm.edu.my
9   Instituto Odontologico das Américas (IOA-Pelotas), 1399—Centro, Pelotas 96020-360, RS, Brazil; irfan_dentart@yahoo.com
10  Department of Periodontology and Implantology, Karnavati School of Dentistry, Karnavati University, 907/A, Uvarsad, Gandhinagar 382422, Gujarat, India; santosh@ksd.ac.in
11  Department of Medical Laboratory Sciences, Faculty of Health Sciences, Eswatini Medical Christian University, P.O. Box A624, Swazi Plaza Mbabane, Mbabane H101, Hhohho, Eswatini; etta5013@gmail.com
12  Department of Pharmacology and Public Health, School of Medicine, American University of Integrative Sciences (AUIS), Bridgetown BB 11114, Barbados; srahman@auis.edu
13  Department of Physiology, Khulna City Medical College and Hospital, 33 KDA Avenue, Hotel Royal Mor, Khulna Sadar, Khulna 9100, Bangladesh; sinhasusmita24@gmail.com
14  Unit of Pharmacology, Faculty of Medicine and Defence Health, Universiti Pertahanan Nasional Malaysia (National Defence University of Malaysia), Kuala Lumpur 57000, Malaysia
*   Correspondence: runurono@gmail.com or mainul@upnm.edu.my; Tel.: +60-109265543

**Abstract:** The measures put in place to contain the rapid spread of COVID-19 infection, such as quarantine, self-isolation, and lockdown, were supportive but have significantly affected the mental wellbeing of individuals. The primary goal of this study was to review the impact of COVID-19 on mental health. An intensive literature search was conducted using PsycINFO, PsyciatryOnline, PubMed, and the China National Knowledge Infrastructure (CNKI) databases. Articles published between January 2020 and June 2022 were retrieved and appraised. Reviews and retrospective studies were excluded. One hundred and twenty-two (122) relevant articles that fulfilled the inclusion criteria were finally selected. A high prevalence of anxiety, depression, insomnia, and post-traumatic stress disorders was reported. Alcohol and substance abuse, domestic violence, stigmatization, and suicidal tendencies have all been identified as direct consequences of lockdown. The eminent risk factors for mental health disorders identified during COVID-19 include fear of infection, history of mental illness, poor financial status, female gender, and alcohol drinking. The protective factors for mental health include higher income levels, public awareness, psychological counseling, social and government support. Overall, the COVID-19 pandemic has caused a number of mental disorders in

addition to economic hardship. This strongly suggests the need to monitor the long-term impact of the COVID-19 pandemic on mental health.

**Keywords:** pandemic; mental health; COVID-19; lockdown; anxiety; depression; stress

## 1. Introduction

The severe acute respiratory syndrome caused by coronavirus 2 (SARS-CoV-2) was first discovered in late 2019 in Hubei Province, Wuhan, China. The infection spread rapidly across the globe, and on March 11, 2020 was named the COVID-19 pandemic by the World Health Organization (WHO) [1–3]. The SARS-CoV-2 and the earlier SARS-CoV-1 are both zoonotic viruses, and evidence suggests that about half of zoonotic viruses are neurotropic because they invade the central nervous system. The neurotropic viruses infect brainstem nuclei, disrupting the regular rhythms and homeostatic control of respiration. During SARS-CoV-1 in 2003 and Middle East Respiratory Syndrome in 2012, many patients exhibited neurotoxic symptoms, leading to neurological and mental disorders [4–8]. However, scientists have yet to establish whether SARS-CoV-2 infection in the brain, in addition to lockdown, causes neurodegenerative or mental disorders. Therefore, there is a need to monitor the long-term impact of SARS-CoV-2 infection in the brain [4–9]. In general, the neurological symptoms of viral infection in the central nervous system (CNS) include delirium, dizziness, loss of smell and taste, headache, loss of consciousness, generalized body weakness, muscle pain, and cerebrovascular complications, Figure 1 [4,5].

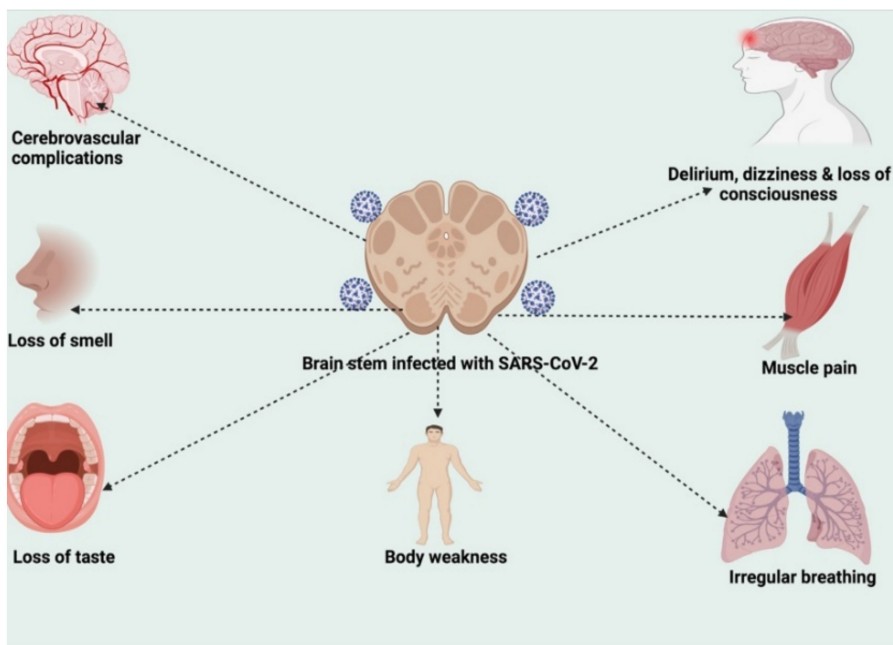

**Figure 1.** Some neurological signs of COVID-19 infection.

The SARS-CoV-2 infection has undoubtedly caused unprecedented morbidity and mortality worldwide. As a result, WHO and countries strategized suitable countermeasures to curb the fast spread of the SARS-CoV-2 infection. These include travel restrictions and the closure of public places such as markets, schools, train stations, airports, seaports, etc. Others are physical distancing, self-isolation, use of facemasks, and hygienic practices like frequent hand washing and hand sanitizer [1,3,9]. Indisputably, these have led to the social and economic shutdown which is very detrimental to the individual's mental health. Several factors were responsible for the association between the COVID-19 pandemic and mental illness. These include anger, hopelessness, sleepless nights, loneliness, and a

significant increase in house chores in the presence of everyone being at home [1,3,9,10]. Mental disorders are highly prevalent and are one of the most neglected diseases worldwide. Common examples of mental illnesses include stress, insomnia, anxiety, depression, and post-traumatic stress disorders (PTSD); see Figure 2. These mental disorders are frequently associated with substance use, and in some cases suicidal tendencies [1,3,6,9–13]. Notably, a number of recent reviews and meta-analyses have reported a high prevalence of mental disorders [14–21].

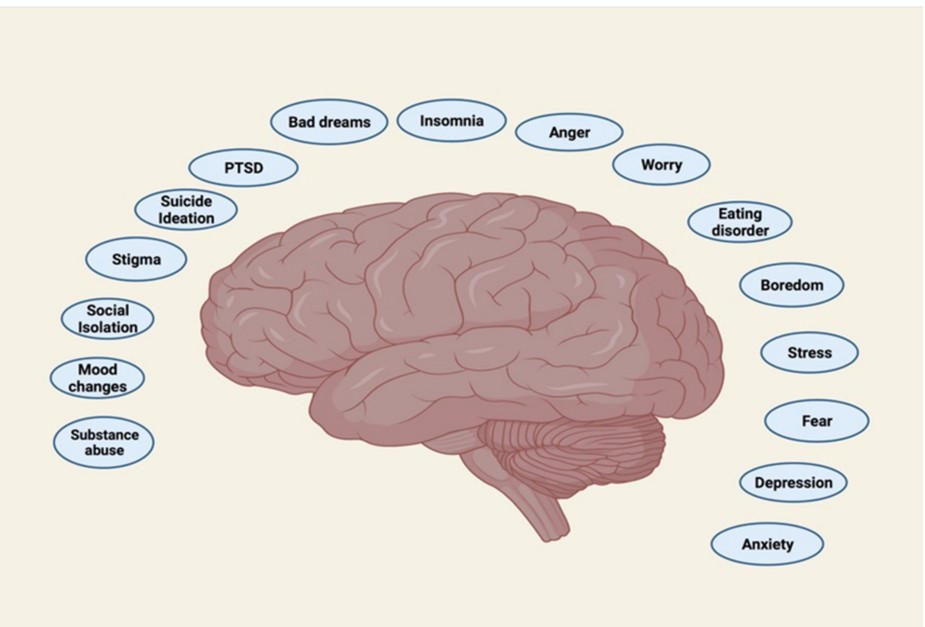

**Figure 2.** Mental disorders and impacts of lockdown associated with COVID-19.

Studies involving the CNS have reported neurological symptoms of COVID-19 infection with different levels of severity. Butowt et al. reported anosmia and ageusia, indicating that the infection has invaded the neurons [22]. Delirium and post-infectious Guillain-Barre syndrome (GBS) were identified as late symptoms [23,24]. Sabel et al. revealed that post-coverage cognitive deficits were also common in some cases [25]. Cao et al. observed that the pro-inflammatory cytokines directly affect the brain, and later thrombogenesis may cause stroke [26]. The overall effect of COVID-19 pandemic on the brain suggests that the virus may produce mental disorders in the long run. The SARS-CoV-2 virus directly attaches to the angiotensin-converting enzyme (ACE) receptor from where it enters a cell and replicates. The host cells then release a suppressed T-cell implying a decreased personal immunity and consequently leading to the CNS invasion [27]. Remarkably, SARS-CoV-2 viruses directly affect sympathetic activity, which decreases serotonin and dopamine concentrations and causes stress [28]. Accordingly, stress directly stimulates the pituitary gland and causes the release of corticotrophin-releasing hormone (CRH) and adrenocorticotropic hormone (ACTH), leading to increased cortisol and vulnerability to further infection [10]. The combined effect of these physiological changes will ultimately cause mental disorders. This study was carried out to investigate the prevalence of mental disorders caused by lockdown, movement restriction, alcohol and substance use.

## 2. Objectives of Study

The first objective of this study is to review published articles on mental disorders due to the COVID-19 pandemic. The second objective is to establish the most commonly reported mental disorders. The third is to establish risk factors for developing mental disorders. The last objective is to establish protective factors for mental disorders.

## 3. Materials and Methods

### 3.1. Search Strategy

Online searches were conducted according to the PRISMA guidelines (Prisma-p, 2015) (Moher et al., 2015) [29]. The first two authors [ARA & MAT] conducted the initial electronic searches using four scientific literature databases, including PsycINFO, PsyciatryOnline, PubMed, and China National Knowledge Infrastructure (CNKI) to obtain the relevant articles. The search terms used include 'COVID-19 pandemic', 'SARS-CoV-2', 'mental health', 'mental disorders', 'psychological disturbance', 'substance use', 'incidence of suicide', 'lockdown', 'quarantine', 'self-isolation', 'stress', 'anxiety', 'depression', 'PTSD', 'insomnia', 'worry', 'fear', 'obsessive-compulsive disorder', and 'eating disorder'.

### 3.2. Study Selection

The authors screened the titles and abstracts of the relevant articles retrieved. In the case of uncertainty, full texts were reviewed. Finally, all authors read the full texts of the eligible studies individually and selected the number of articles for the final review. A manual search of the reference sections of the suitable papers was conducted to identify studies not found through the database searches. This review included preprinted articles where necessary because research on the COVID-19 pandemic is a novel area of study. The quality of the articles retrieved were examined using the Newcastle Ottawa Scale except for the preprinted articles [30,31]. The quality of the pre-printed manuscripts was assessed based on their study design, the instrument used, the sample size, and the track record of the authors. In the course of this review, certain terms were used interchangeably, such as 'sleep disturbance' and 'insomnia'; 'stress' and 'distress'; 'psychological distress' and 'psychological disturbances'; 'post-traumatic stress disorders' and 'post-traumatic and related disorders'. The article retrieval, screening, and inclusion flow chart is shown in Figure 3 [32–38].

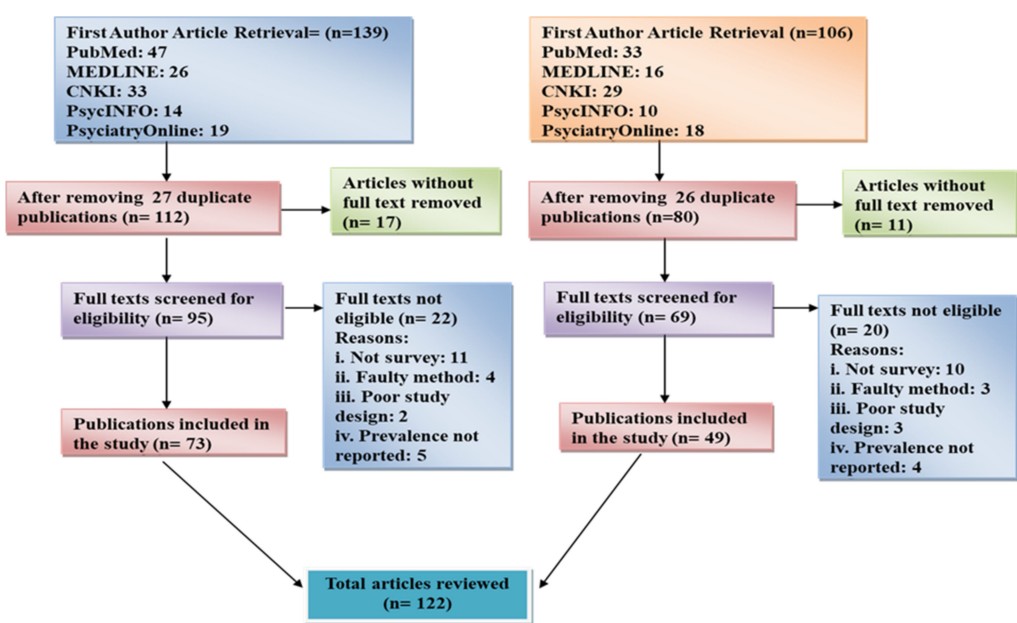

**Figure 3.** Articles retrieval and screening flowchart.

### 3.3. Inclusion and Exclusion Criteria

Inclusion Criteria: i. Original studies. ii. Studies published between January 2020 and June 2022. iii. Quantitative studies. iv. Studies published in the English language. Exclusion Criteria: i. Retrospective studies. ii. Studies that didn't focus on the prevalence of mental health. iii. Review articles.

*3.4. Data Synthesis*

Initially, 206 articles were independently retrieved from the selected databases by the first two authors (Figure 3). In addition, 39 more articles were obtained through the manual search by reading the reference sections of the first sets of articles, making a total of 245 articles. After cross-checking the retrieved articles, a total of 53 duplicate articles were screened and removed. An additional 28 articles were excluded because they don't have full text. Subsequently, 42 articles were excluded because they were not surveys, had a faulty method, or had poor study design. Some of the excluded studies were conducted either through verbal interviews or focused group discussion; others did not report the prevalence of mental disorders, and only abstracts were available among the rest. Finally, 122 published studies that met the inclusion criteria were reviewed (Table 1). Studies were grouped under the most relevant subheadings; however, there was an overlap in some studies (Table 2).

**Table 1.** Surveys Outcomes Showing the Prevalence, Risks, and Protective Factors to Mental Health Disorders.

| S/N | Study | Country | Study Population | Findings | | |
|-----|-------|---------|------------------|----------|---|---|
| | | | | Prevalence (%) | Risk Factors | Protective Factors |
| 1. | Dawel et al., 2020 [39] | Australia | 1296 | i. Generalized Anxiety Disorder (16%) <br> ii. Major Depressive Disorder (20%) | i. Financial distress <br> ii. Loss of job | Government support |
| 2. | Li et al., 2021 [34] | Australia | 760 | i. Anxiety (40%) <br> ii. Psychological distress (48%) <br> iii. Sleep disturbance (41%) | History of mental health | i. Mental health support <br> ii Government support |
| 3. | Wilson et al., 2022 [40] | Australia | 555 | i. Anxiety/Depression (mild, 85%) <br> ii. Alcohol Use (Moderate, 80%) | i. Unemployment <br> ii. Financial difficulties <br> iii. Reduced accessibility to hobbies | i. Avoid distress, <br> ii. Do things differently |
| 4. | Simon et al., 2021 [41] | Austria | 560 | i. Anxiety (16%) <br> ii. Depression (11%) | i. History of mental health <br> ii. Wellbeing reduction | i. Mental health support <br> ii. Social support |
| 5. | Jassim et al., 2021 [42] | Bahrain | 502 | i. Depression (40%) <br> ii. PTSD (20%) <br> iii. Perceived stigma (53%) | i. Female gender <br> ii. History of mental health issues <br> iii. Young adult | Psychological interventions |
| 6. | Islam et al., 2021 [43] | Bangladesh | 975 | i. Anxiety (5%) <br> ii. Poor sleep (44–55%) <br> iii. Fear (59%) | i. Female gender <br> ii. Fear of infection <br> iii. Poor income <br> iv. Poor physical illness | i. Online counseling <br> ii. Government support |
| 7. | Das et al., 2021 [44] | Bangladesh | 672 | i. Anxiety (64%) <br> ii. Depression (38%) <br> iii. Insomnia (73%) | i. Female sex <br> ii. Unemployment <br> iii. Being a student <br> iv. Obesity <br> v. Living without a family | Supportive programs |
| 8. | Islam et al., 2020 [45] | Bangladesh | 475 | i. Anxiety (18%) <br> ii. Depression (15%) | i. Living with families <br> ii. Being a student | i. Online classes <br> ii. Government support |
| 9. | Mehareen et al., 2021 [46] | Bangladesh | 333 | i. Anxiety (Public University 54%, Private University 33%) <br> ii. Depression (Public university 59%, Private University 31%) | i. Female gender <br> ii. Level of study <br> iii. Nuclear families | i. Psychological interventions <br> ii. Government support |

| S/N | Study | Country | Study Population | Findings | | |
|-----|-------|---------|------------------|----------|---|---|
| | | | | Prevalence (%) | Risk Factors | Protective Factors |
| 10. | Lopes et al., 2021 [47] | Brazil | 1224 | i. Anxiety (53%)<br>ii. Depression (61%)<br>iii. Stress (58%) | i. Female gender<br>ii. Younger age<br>iii. Having a chronic diseases | i. Educational actions<br>ii. Increasing psychological wellbeing |
| 11. | Gadermann et al., 2021 [3] | Canada | 3000 | i. Deteriorated mental health (44.3%)<br>ii. Anxiety and worry (52%)<br>iii. Suicidal thoughts (8%) | i. Having children <18 years<br>ii. Alcohol consumption | i. Free digital technologies<br>ii Government supports |
| 12. | Maximova et al., 2021 [48] | Canada | 1095 | i. Boredom (Girls 48%, Boys 36%)<br>ii. Trouble paying attention (Girls 36%, Boys 39%) | | Playing video games |
| 13. | Song et al., 2020 [49] | China | 14,825 | i. Depression (25%)<br>ii. PTSD (9%) | i. Male gender<br>ii. Old age<br>iii. Working in Hubei province<br>iv. Low social support | i. Psychological interventions<br>ii. Mental health promotion |
| 14. | Cao et al., 2020 [50] | China | 7143 | i. Severe Anxiety (1%)<br>ii. Moderate Anxiety (3%)<br>iii. Mild Anxiety (21%) | i. Having infected acquaintances<br>ii. Worry about economy<br>iii. Worry about school | i. Living in a city<br>ii. Higher level of income<br>iii. Living with parents<br>iv. Government support |
| 15. | Li et al., 2021b [51] | China | 7090 | i. Anxiety (19%)<br>ii. Depression (21%)<br>iii. Poor self-rated health (10%) | Fear of infection<br>Work intensity | Improve the working condition |
| 16. | Huang et al., 2020 [52] | China | 6261 | i. Anxiety (Moderate 14%, Severe 5%)<br>ii. Depression (Moderate 17%, Severe 8%) | i. Being single<br>ii. People from Hubei province<br>ii. Infected people | Psychological intervention |
| 17. | Zhu et al., 2020 [53] | China | 5062 | i. Anxiety (24%)<br>ii. Depression (14%)<br>iii. Stress (30%) | i. Female gender<br>ii. Chronic diseases<br>iii. Fear of infection<br>iv. History of mental disorders | i. Psychological support<br>ii. Government support |
| 18. | Liu et al. 2020a [54] | China | 4679 | i. Anxiety (16%)<br>ii. Depression (35%)<br>iii. Psychological distress (16%) | i. Divorce/widow<br>ii. Younger age<br>iii. Nurse<br>iv. Not living with family | psychiatric interventions |

<table>
<tr><td colspan="2" align="center">**Table 1.** *Cont.*</td></tr>
</table>

| S/N | Study | Country | Study Population | Findings | | |
|-----|-------|---------|------------------|----------|---|---|
| | | | | **Prevalence (%)** | **Risk Factors** | **Protective Factors** |
| 19. | Ren et al., 2020 [55] | China | 3600 | i. Anxiety (Mild 19%, Moderate 5%, Severe 1%). <br>ii. Depression (Mild 17%, Moderate 4%, Severe 1%) | i.Surgical nurses <br>ii. Divorce/widowed <br>iii. Care for COVID-19 patients | Mental health support |
| 20. | Duan et al., 2020 [56] | China | 3254 | i. Anxiety (31%) <br>ii. Depression (22%) | i. Resident in Hubei province <br>ii. Infected family member <br>ii. Internet addiction <br>iii. Old age | I. Psychological interventions <br>ii. Conducting research |
| 21. | Huang et al., 2021 [57] | China | 3113 | i. Anxiety (13%) <br>ii. Depression (15%) <br>iii. Stress (7%) | i. Smoking <br>ii. Alcohol drinking | i. Family support <br>ii. Psycho intervention |
| 22. | Hou et al., 2020 [58] | China | 3063 | i. Anxiety (13%) <br>ii. Depression (14%) <br>iii. Stress (7%) | i. Female gender <br>ii. Old age <br>iii. Unemployment <br>iv. Exposure to COVID-19 news | i. Limit exposure to social media <br>ii. Mental health prevention |
| 23. | Cai et al., 2020 [59] | China | 2346 | i. Anxiety (Frontline workers 16%, Non-Frontline workers 7%) <br>ii. Depression (Frontline workers 14%, Non-Frontline workers 10%) <br>iii. Insomnia (Frontline workers 47%, Non-Frontline workers 29%) <br>iv. Suicidal ideation (Frontline workers 12%, Non-Frontline workers 9%) | i. Frontline worker <br>ii. Working in Wuhan | i. Mental health support |
| 24. | Lu et al., 2020a [60] | China | 2299 | i. Anxiety (Moderate 26%, Severe 3%) <br>ii. Depression (Mild 12%, Moderate 0.3%) | i. Healthcare workers <br>ii. Working in ICU | Improving the mental health |
| 25. | Que et al., 2020 [61] | China | 2285 | i. Anxiety 46%, <br>ii. Depression 45%, <br>iii. Insomnia 29% | Front-line healthcare | i. Timely interventions <br>ii. Proper information feedback. |

**Table 1.** *Cont.*

| S/N | Study | Country | Study Population | Findings | | |
|---|---|---|---|---|---|---|
| | | | | Prevalence (%) | Risk Factors | Protective Factors |
| 26. | Zhang et al., 2020a [62] | China | 2182 | Medical vs. Nonmedical Workers<br>i. Anxiety (13% vs. 9%)<br>ii. Depression (12% vs. 10%)<br>iii. Insomnia (38% vs. 31%)<br>iv. OCD (5% vs. 2%) | I. Health worker<br>ii. Organic disease<br>iii. Living in rural area | Recovery programs |
| 27. | Liu et al., 2020b [63] | China | 2031 | i. Anxiety (18%)<br>ii. Depression (15%)<br>iii. Stress (10%) | i. Health worker<br>ii. Older age<br>iii. Working in frontline | Psychological crisis interventions |
| 28. | Wang et al., 2020a [64] | China | 1738 | i. Anxiety (29%)<br>ii. Depression (17%)<br>iii. Stress (8%) | i. Physical symptoms<br>ii. Low knowledge about the infection | Government Financial support |
| 29. | Wang et al., 2020b [65] | China | 1599 | i. Feel nervous (57%)<br>ii. Bad dreams (38%)<br>iii. Emotional disturbances (48%) | i. Unmarried<br>ii. Younger age<br>iii. History of the visit to Wuhan | Psychological interventions |
| 30. | Lai et al., 2020 [32] | China | 1257 | i. Anxiety (45%)<br>ii. Depression (50%)<br>iii. Distress (72%)<br>iv. Insomnia (34%) | i. Female gender<br>ii. Nurses<br>iii. Frontline health care workers<br>iv. Working in Wuhan | i. Mental health intervention<br>ii. Special attention to women and nurses |
| 31. | Guo et al., 2021 [33] | China | 1091 | i. Anxiety (53%)<br>ii. Depression (56%)<br>iii. PTSD (11%)<br>iv. Insomnia (79%) | iii. Having higher degrees<br>iv. Working in Wuhan | Early mental health intervention |
| 32. | Kang et al., 2020 [66] | China | 994 | Mild psychological disturbance (34%)<br>ii. Moderate psychological disturbance (22%)<br>iii. Severe psychological disturbance and (6%) | i. Low access to mental healthcare<br>ii. Dealing with confirmed cases | Mental health interventions |
| 33. | Du et al., 2020 [67] | China | 687 | i. Anxiety (30%)<br>ii. Depression (18%)<br>iii. Stress (14%) | i. Female gender<br>ii. Healthcare worker<br>ii. Medical students | i. Preventive measures<br>ii. Active coping strategies |

**Table 1.** *Cont.*

| S/N | Study | Country | Study Population | Findings | | |
|-----|-------|---------|------------------|----------|---|---|
| | | | | Prevalence (%) | Risk Factors | Protective Factors |
| 34. | Ning et al., 2020 [68] | China | 612 | i. Anxiety (Neurological nurses (20%, Doctors 13%) ii. Depression (Neurological nurses 30%, 20%) | i. Female gender ii. Nurses iii. Younger age iv. Junior Health worker | i. Provision of PPE ii. Psychological assistance. |
| 35. | Liang et al., 2020 [69] | China | 584 | i. Psychological problems (40%) ii. PTSD (14%) | i. Low level of education ii. Employment status iii. Marital status | i. Government support ii. Psychological counseling |
| 36. | Liu et al., 2020c [70] | China | 512 | i. Mild Anxiety (10%), ii. Moderate Anxiety (1.4%) ii. Severe Anxiety (0.8%). | i. Working in Hubei province ii. Direct contact treating infected patients | i. Psychological support ii. Government support |
| 37. | Juan et al., 2020 [71] | China | 456 | i. Anxiety (32%) ii. Depression (30%) iii. Stress (43%) iv. Psychological distress | i. Female gender ii. Low income iii. Younger adults iv. Fear of infecting others | i. Social support ii. Psychological intervention |
| 38. | Zhang et al., 2020b [72] | China | 263 | Apprehension (52%) | Old age | i. Family Support ii. Attention to mental health |
| 39. | Liu et al., 2020d [73] | China | 217 | i. Anxiety (Male 20%, Female 24%), ii. Depression (Male 30%, Female 39%) | i. Female gender ii. Living in Hubei Province ii. Level in school | Effective screening procedures |
| 40. | Rodriguez-Hidalgo et al., 2020 [74] | Ecuador | 640 | i. Anxiety (60%) ii. Depression (80%) | i. Female gender ii. Fear of infection | i. Psychological training ii. Counseling program |
| 41. | Deek et al., 2021 [75] | Egypt, Lebanon, Libya, Saudi Arabia, Sudan | 2783 | i. Anxiety (3–8%) ii. Depression (2–7%) iii. Insomnia (2–9%) | i. Poverty ii. Change of Government | Government support |
| 42. | Herbert et al., 2021 [76] | Egypt, Germany | 220 | i. Anxiety (50%) ii. Depression (52%) Depressive symptoms (65.5%) | i. Worries about health ii. Difficulties in identifying feelings iii. Difficulties in learning behavior | Psychological interventions |
| 43. | Fancourt et al., 2020 [77] | England | 36,520 | i. Anxiety (Moderate 12%, Severe 10%) ii. Depression (Moderate 13%, Severe 8%) | i. Female gender ii. Lower education iii. Younger adults iv. Existing mental illness | Mental health support |
| 44. | Zaninotto et al., 2021 [78] | England | 5146 | i. Anxiety (9–11%) ii. Depression (23–29%) | i. Women ii. Being Single iii. Pre-existing health issues iv. Poor economic status | i. Mental health screening ii. Psychological support |

| S/N | Study | Country | Study Population | Findings | | |
|---|---|---|---|---|---|---|
| | | | | Prevalence (%) | Risk Factors | Protective Factors |
| 45. | Assefa et al., 2021 [79] | Ethiopia | 710 | i. Anxiety (35%) ii. Depression (30%) iii. Stress (38%) | i. Married ii. Old age iii. Low level of education iv. History of mental disorders | i. Psychological counseling ii. Coping strategies |
| 46. | Girma et al.,2021 [80] | Ethiopia | 610 | i. Moderate stress (68%) ii. Severe stress (14%) | i. Large family size ii. Chronic diseases iii. Old age | i. Prevention of psychological impacts of COVID-19 ii. Mental health counseling |
| 47. | Geweniger et al., 2022 [81] | Germany | 1619 | i. Children mental health problems (57%) ii. Parent depression (31%) | i. Low socioeconomic status ii. Complex chronic disease iii. Parents with depression | Political measures to help children |
| 48. | Schäfer et al., 2020 [82] | Germany | 1591 | i. Psychopathological symptoms (10%) ii. PTSD (15%) | i. Younger age ii. Female gender | Social support |
| 49. | Rek et al., 2021 [37] | Germany | 511 | i. Anxiety (11%) ii. Depression (24%) ii. PTSD (5%) iii. Substance use (1%) iv. Eating disorder (4%) | i. Political restriction ii. Existing psychiatric illness iii. Conspiracy beliefs | Self-assessment |
| 50. | Knolle et al., 2021 [83] | Germany, UK | 782 | i. Psychological symptoms, Germany, UK (25%) ii. Depression, Germany, UK (20–50%) | i. High consumption of Marijuana ii. Use of social media | i. Being older, ii. Having a better education |
| 51. | Magklara et al., 2020 [84] | Greece | 1232 | i. Sleep problems (8%) ii. Stress (6%) | i. Mental health history ii. Unemployment iii. Family conflict | Public health policies |
| 52. | Reddy et al., 2020 [85] | India | 891 | i. Anxiety (15%) ii. Depression (22%) iii. Anxiety and Depression (28%) | i. Being single ii. Worries regarding school opening iii. Online teaching | Timely Psychological intervention. |
| 53. | Saraswathi et al., 2020 [86] | India | 217 | i. Anxiety (33%) ii. Depression (36%) iii. Stress (25%) | Direct contact with COVID-19 patients | Mental health intervention |

Table 1. *Cont.*

| S/N | Study | Country | Study Population | Findings | | |
|-----|-------|---------|------------------|----------|---|---|
| | | | | Prevalence (%) | Risk Factors | Protective Factors |
| 54. | Zukhra et al., 2021 [87] | Indonesia | 247 | i. Mild Anxiety (30%) ii. Moderate Anxiety (5%) iii. Severe Anxiety (0.4%) | i. Female gender ii. Younger age iii. Living in COVID-19 red zone | i. Psychological support ii. Mental health counseling |
| 55. | Sharif Nia et al., 2021 [88] | Iran | 70,180 | i. Anxiety (Moderate 21%, Severe 59%) ii. Depression (Mild 18%, Moderate 18%) iii. Stress (Moderate 59%, Severe 7%) | i. Female gender ii. Married iii. Level of education | Psychological interventions |
| 56. | Shahriarirad et al., 2021 [12] | Iran | 8591 | i. Anxiety (20%) ii. Depression (15%) | i. Female gender ii. Healthcare worker | i. Older age ii. Being Married iii. Getting information from medical journals |
| 57. | Azizi et al., 2021 [36] | Iran | 7626 | i. Anxiety (43%) ii. Depression (45%) iii. Stress (35%) | i. Female gender ii. Younger age iii. Physical illness iv. History of mental disorders | i. Psychological screening ii. Government support |
| 58. | Hassannia et al., 2020 [89] | Iran | 2045 | i. Anxiety (66%) ii. Depression (42%) iii. Stress (35%) | i. Female gender ii. Younger age iii. Doctors and nurses iv. Infected individuals | i. Psychological intervention ii. Helping vulnerable people |
| 59. | Salehian et al., 2021 [90] | Iran | 1910 | i. Anxiety (40%) ii. Depression (22%) iii. PTSD (62%) | i. Women, ii. Younger age iii. Divorced/widowed iv. History of psychiatric disorders | Continuous monitoring of the psychological consequences of corvid-19 |
| 60. | Mani et al., 2020 [91] | Iran | 922 | i. Anxiety (19%) ii. Depression (6%) | i. Old age ii. Female gender iii. Lack of trust in Government | Government support |
| 61. | Kausar et al., 2021 [92] | Iran | 500 | i. Anxiety (Mild 11%, Moderate 13%) ii. Depression (Mild 18%, Moderate 18%) iii. Stress (Mild 11%, Moderate 4%) | | Counseling services |

Table 1. *Cont.*

| S/N | Study | Country | Study Population | Findings | | |
|-----|-------|---------|------------------|----------|---|---|
| | | | | Prevalence (%) | Risk Factors | Protective Factors |
| 62. | Chen et al., 2021 [93] | Iran | 474 | i. Anxiety (43%)<br>ii. Depression (45%)<br>iii. Stress (35%) | i. Old age<br>ii. Female gender<br>iii. Chronic diseases | Social support |
| 63. | Mohammadi et al., 2020 [94] | Iran | 462 | i. Anxiety (General population 96%, COVID-19 patients 98%)<br>ii. Depression (General population 52%, COVID-19 patients 54%)<br>iii. Stress (General population 49%, COVID-19 patients 47%) | i. Female gender<br>ii. Younger age<br>iii. Comorbidity diseases | Psychological interventions |
| 64. | Orgilés et al., 2020 [95] | Italy, Spain | 1114 | i. Anxiety (28%)<br>ii. Worry (30%)<br>iii. Stress (Children 11%, Parents 35%) | Fear of infection | Use of mobile phones and computers |
| 65. | Rossi et al., 2020 [96] | Italy | 18,147 | i. Anxiety (21%),<br>ii. Depression (17%)<br>iii. Stress (22%)<br>iv. PTSD (37%)<br>v. Insomnia (7%) | i. Female Gender<br>ii. Younger age<br>iii. Quarantine | Monitoring of the mental health status |
| 66. | Davico et al., 2021 [97] | Italy | 2419 | i. Psychological impact (33%)<br>ii. PTSD (31%) | i. Fear of infection<br>ii. Home confinement | Physiological intervention |
| 67. | Villani et al., 2021 [98] | Italy | 501 | i. Anxiety (35%)<br>ii. Depression (73%) | i. Female gender<br>ii. Students<br>iii. Inability to see partner | Physical activity |
| 68. | Giusti et al., 2020 [99] | Italy | 330 | i. Anxiety (31%)<br>ii. Depression (27%)<br>iii. Stress (34%)<br>iv. PTSD (37%) | i. Female gender,<br>ii. Being a nurse,<br>iii. Contact with COVID-19 patients | Monitoring and timely treatment |
| 69. | Levkovich and Shinan-Altman, 2021 [100] | Israel | 1407 | i. Anxiety and Worry (40%)<br>ii. High level of fear (20%) | i. Fears of infection<br>ii. Adjustment to the new reality | Government intervention |
| 70. | Basheti et al., 2021 [101] | Jordan | 450 | i. Anxiety (34%)<br>ii. Depression (26%) | i. Smoking<br>ii. Low income | Government intervention |
| 71. | Shah et al., 2021 [11] | Kenya | 433 | i. Anxiety (44%)<br>ii. Depression (54%)<br>iii. Insomnia (41%) | i. Hospital workers<br>ii. Female gender | i. Government support<br>ii. Doctors' welfare |

**Table 1.** *Cont.*

| S/N | Study | Country | Study Population | Findings | | |
|---|---|---|---|---|---|---|
| | | | | Prevalence (%) | Risk Factors | Protective Factors |
| 72. | Wong et al., 2021 [35] | Malaysia | 1163 | i. Anxiety (55%) ii. Depression (59%) iii. Stress (31%) | i. Young people ii. Females iii. Poor financial conditions | i. Psychological counselling ii. Government support |
| 73. | Bahar Moni et al., 2021 [102] | Malaysia | 720 | i. Moderate psychological distress (62%) ii. High levels of fear (27%) | i. Alcohol drinking ii. Fear of infection iii. Care of COVID-19 patient iii. Poor financial situation | Behavioral interventions |
| 74. | Sundarasen et al., 2020 [103] | Malaysia | 983 | i. Mild to moderate anxiety (20%) ii. Severe anxiety (7%) iii. Extreme anxiety (3%) | i. Financial constraints ii. Remote online teaching | i. Mental health support ii. Government support |
| 75. | Baloch et al., 2021 [104] | Malaysia | 494 | i. Mild to moderate anxiety (25%) ii. Severe anxiety (9%) iii. Extreme anxiety (7%) | i. Online teaching ii. Uncertainty about their academic performance | Mental health interventions |
| 76. | Norhayati et al., 2021 [105] | Malaysia | 306 | Depressive symptoms (Frontline healthcare 28%, Non-frontline healthcare 38%) | | Psychological support |
| 77. | Chinna et al., 2021 [106] | Malaysia, Saudi Arabia, Pakistan, Bangladesh, China, India, and Indonesia | 3679 | i. Mild to moderate anxiety (22%) ii. Severe anxiety (14%) | i. Female gender ii. Substance use | i. Social Support ii. Government support |
| 78. | Cortés-Álvarez et al., 2021 [107] | Mexico | 1105 | i. Moderate-severe depression (16%) ii. Moderate-severe anxiety (23%) iii. Moderate-severe stress (20%) | i. Female gender ii. Older age iii. Contact with a confirmed case | i. Hand hygiene ii. Wearing masks |
| 79. | Khanal et al., 2021 [108] | Nepal | 475 | i. Anxiety (33%) ii. Insomnia (7%) | i. Nurses ii. family members with chronic diseases ii. stigma | i. Monitor the psychological illness ii. Psychological intervention |
| 80. | Khanal et al., 2020 [13] | Nepal | 475 | i. Anxiety (42) ii. Depression (38%) iii. Insomnia (34%) | i. History of mental health problems ii. Stigma | i. Government support system ii. Availability of PPE |

**Table 1.** *Cont.*

| S/N | Study | Country | Study Population | Findings | | |
|---|---|---|---|---|---|---|
| | | | | Prevalence (%) | Risk Factors | Protective Factors |
| 81. | Van der Goot et al., 2021 [109] | Netherland | 259 | i. Mild Psychological distress (28–50%) ii. Moderate Psychological distress (7–20%) iii. Severe Psychological distress(13–30%) | | Psychological support |
| 82. | Tobin et al., 2021 [110] | Nigeria | 543 | i. Anxiety (24%) ii. Depression (17%) | i. Female gender ii. Alcohol use iii. Currently on medication | Psychological support |
| 83. | Olaseni et al., 2020 [111] | Nigeria | 502 | i. Anxiety (49–51%), ii. Depression (Males 7–12%, Females 5–14%) iii. Moderate PTSD (Males 18–22%, Females 19–29%) | i. Female gender ii. Increase in number reported cases | Government support |
| 84. | Durowade et al., 2021 [112] | Nigeria | 335 | Psychological effects (84%) | i. Diabetes, asthma, cancers ii. Contact with a confirmed case | i. Public awareness, ii. Subsidizing PPEs iii. Financial stimulus |
| 85. | Adewale et al., 2021 [113] | Nigeria | 322 | i. Severe anxiety (6%) ii. Severe depression (3%) iii. Severe psychological impact (20%) | i. Increase in time spent on social media and TV ii. Decrease in physical activity | i. Psychosocial support ii. Government support |
| 86. | Fadipe et al., 2021 [114] | Nigeria | 160 | i. Depression (28%) ii. Anxiety (28%) iii. Suicidal ideation (4%) | i. Fear of infecting ii. Employment status iii. History of negative emotion | Nigeria |
| 87. | Afolabi, 2020 [115] | Nigeria | 132 | i. Poor mental wellbeing (55%) ii. Worries (71%) | Sleeplessness | i. Government support ii. Social support |
| 88. | Khamis et al., 2020 [116] | Oman | 402 | i. Mild Anxiety 40% ii. Moderate Anxiety 19% iii. Severe Anxiety 9% iv. Poor sleep 39% | i. Care for COVID-19 patients ii. Being a citizen | Mental health support |
| 89. | Hayat et al., 2021 [117] | Pakistan | 1094 | i. Anxiety (Moderate to Severe 33%) ii. Depression (Mild 45%, Moderate 12%) | i. Female gender ii. Old age iii. Married | i. Psychotherapy ii. Counselling services |

| S/N | Study | Country | Study Population | Findings | | |
|---|---|---|---|---|---|---|
| | | | | **Prevalence (%)** | **Risk Factors** | **Protective Factors** |
| 90. | Majeed and Ashraf, 2020 [118] | Pakistan | 63 | i. Anxiety (60%)<br>ii. Fear (70%) | i. Uncertainty<br>ii. Misinformation<br>iii. Social distancing/isolation | i. Psychosocial interventions<br>ii. Government support |
| 91. | Radwan et al., 2021 [119] | Palestine | 420 | i. Anxiety (Mild 1.6%, Severe 12%)<br>ii. Depression (Mild%, Severe 9%)<br>iii. Stress (Mild 12%, Severe 13%) | i. Female gender<br>ii. Family poor income<br>iii. large Family size<br>iv. Younger age | Mental health support |
| 92. | Villarreal-Zegarra et al., 2021 [120] | Peru | 830 | i. Depression (16%)<br>ii. Anxiety (12%)<br>iii. PTSD (15%) | i. Healthcare workers<br>ii. Fear infection | i. Preventive actions<br>ii. Surveillance of mental health |
| 93. | Stack et al., 2020 [121] | Poland | 36 | Substance use (17–52%) | Availability of substances | Government support |
| 94. | Karpenko et al., 2020 [122] | Russia | 352 | i. Anxiety (30%)<br>ii. Depression (17) | i. Fear of infection<br>ii. Self-isolation<br>iii. Fear of financial problems | i. Mental health support<br>ii. Social support |
| 95. | Alkhamees et al., 2020 [123] | Saudi Arabia | 1160 | i. Moderate to severe anxiety (24%)<br>ii. Moderate to severe depression (28%)<br>iii. Moderate to severe stress (22%) | i. Female gender<br>ii. High-school students<br>iii. Healthcare workers | Psychological interventions |
| 96. | Al-Rahimi et al., 2021 [38] | Saudi Arabia | 1030 | i. Anxiety (21%)<br>ii. Worrying thoughts (20%) | i. Female gender<br>ii. Lower education<br>iii. Middle-aged<br>iv. Divorced or widowed<br>v. Chronic diseases | Psychological interventions |
| 97. | Alyoubi et al., 2021 [124] | Saudi Arabia | 582 | i. Anxiety (22%)<br>ii. Depression (25%)<br>iii. Stress (18%) | i. Pre-existing mental health condition<br>ii. Learning difficulties<br>iii. Insomnia | i. Psychological interventions<br>ii. Government support |
| 98. | Odriozola-González et al., 2020 [125] | Spain | 2530 | i. Anxiety (21%)<br>ii. Depression (34%)<br>iii. Stress (28%) | i. Course of study<br>ii. Year of study | i. Self-isolation<br>ii. Social distancing |

**Table 1.** *Cont.*

| S/N | Study | Country | Study Population | Findings | | |
|-----|-------|---------|------------------|----------|---|---|
| | | | | Prevalence (%) | Risk Factors | Protective Factors |
| 99. | Muñoz-Violent et al., 2021 [126] | Spain | 996 | i. Anxiety (39%)<br>ii. Depression (12%) | i. Female gender<br>ii. Large family size<br>iii. History of mental illness<br>iv. Fear of infection | Coping skills |
| 100. | Visser and Wyk, 2021 [127] | South Africa | 5074 | i. Anxiety (46%)<br>ii. Depression (35%) | Fear of infection | Psychological interventions |
| 101. | Posel et al., 2021 [128] | South Africa | 2213 | Depression (24%) | Job loss | i. Mental health interventions<br>ii. Re-employment |
| 102. | Werling et al., 2022 [129] | Switzerland | | i. Anxiety (Severe, 33.6%)<br>ii. Depression (Moderate, 44.3%)<br>iii. Stress (Moderate, 50.8%) | i. Loneliness/isolation of the child<br>ii. Worry about child's education<br>iii. Increased media use<br>iv. Missing recreational activities | i. Adequate medical supply<br>ii. Support for families |
| 103. | Krifa et al., 2022 [130] | Tunisia | 366 | i. Anxiety (Severe, 33.6%)<br>ii. Depression (Moderate, 44.3%)<br>iii. Stress (Moderate, 50.8%) | i. Fear of infection<br>ii. Examination stress<br>iii. Low response to students' needs | i. Social support<br>ii. Psychological support<br>iii. Counseling |
| 104. | Al Dhaheri et al., 2021 [131] | United Arab Emirates | 6142 | i. Psychological Distress (31%)<br>ii. Felt horrified (62%)<br>iii. Stress (60%) | i. Female gender<br>ii. Young adults | Support from family |
| 105. | Saddik et al., 2021 [132] | United Arab Emirates | 481 | i. Anxiety (Mild 66%, Severe 32%)<br>ii. Psychological distress (Mild 49%, Severe 37%) | i. Worry about COVID-19<br>ii. Being isolated<br>iii. Contracting COVID-19<br>iv. Feeling stigmatized | i. Mental health preventive policies<br>ii. Psychological support |
| 106. | O'Connor et al. 2022 [133] | UK | 3077 | i. Anxiety (17–22%)<br>ii. Suicidal ideation (13–14%)<br>iii. Depression (23–26%) | i. Female gender<br>ii. Younger age<br>iii. Pre-existing mental health | Psychological interventions |
| 107. | Niedzwiedz et al., 2021 [134] | UK | 9748 | Psychological distress (31%) | Lockdown measures | i. Psychological support,<br>ii. Access to mental health services |
| 108. | Chen and Lucock, 2022 [135] | UK | 1178 | i. Anxiety (50%)<br>ii. Depression (50%) | i. Low exercising<br>ii. High tobacco use<br>iii. Financial concerns<br>iv. Worse personal relations<br>v. Cancellation of an event | i. Social support<br>ii. Psychological therapy<br>iii. Counseling |

| S/N | Study | Country | Study Population | Findings | | |
|-----|-------|---------|------------------|----------|---|---|
| | | | | Prevalence (%) | Risk Factors | Protective Factors |
| 109. | Morgül et al., 2020 [136] | UK | 927 | i. Anxiety (45%) ii. Worried (52%), iii. Angry (49%), | i. Impact of the quarantine ii. Children's screen use time iii. Physical activity | Development of intervention programs |
| 110. | Prasad et al., 2021 [137] | USA | 20,947 | i. Anxiety or depression (38%) ii. Burnout (49%) | i. Fear of exposure ii. Female gender iii. Black race and Latino | Government support |
| 111. | Czeisler et al., 2020 [138] | USA | 5412 | i. Anxiety or Depression (31%) ii. PTSD (26%) iii. Substance use (13%) iv. Suicide tendency (11%) | i. Young adult ii. Ethnic minority iii. Pre-existing psychiatric conditions iv. Unpaid caregivers | i. Community intervention ii. Government support |
| 112. | Czeisler et al., 2021 [139] | USA | 5186 | i. Anxiety or Depression (33%) ii. PTSD (30%) iii. Substance use (15%) iv. Suicide tendency (12%) | i. Wrong coping strategy ii. Employment status iii. History of mental illness | Government support |
| 113. | Vahia et al., 2020 [140] | USA | 3840 | i. Anxiety disorder (6%) ii. Depressive disorder (6%) iii. PTSD (9%) | i. Isolation ii. longer-term physical and financial wellbeing | i. Utilizing technology to maintain contact ii. Mental health services |
| 114. | Dickey-Chasins et al., 2022 [141] | USA | 3006 | Anxiety/depressive (Moderate, 29.1%) | i. Females gender ii. Democrats iii. Sexual minorities iv. Unemployment v. Single/unmarried | i. Social support ii. Government Intervention |
| 115. | Browning et al., 2021 [142] | USA | 2534 | i. Anxiety (22%) ii. Depression (25%) iii. Stress (18%) | i. Female gender ii. Younger age iii. Comorbidity diseases iv. Poor income | i. Mental health support ii. Educational support |
| 116. | Lopez-Castro et al., 2021 [143] | USA | 909 | i. Depression (90%) ii. Anxiety (66%) iii. PTSD (5%) | i. History of infection ii. Emotional Health issues ii. Poor wellbeing | Social support |
| 117. | Son et al., 2020 [144] | USA | 195 | i. Anxiety and Stress (71%) ii. Depressive thoughts (44%) | i. Disruptions of sleeping ii. Fear of infection iii. Decreased social interactions | i. Mental health counseling ii. Self-Management iii. Seeking support |
| 118. | Hamm et al., 2020 [145] | USA | 73 | i. Anxiety (75%) ii. Depression (44%) iii. Social Isolation (36%) | i. Fear of losing the job ii. Financial problems | i. Internet surfing ii. Avoid negative emotions ii. Exercises |

**Table 1.** *Cont.*

| S/N | Study | Country | Study Population | Findings | | |
|-----|-------|---------|------------------|----------|---|---|
| | | | | **Prevalence (%)** | **Risk Factors** | **Protective Factors** |
| 119. | Jow et al., 2022 [146] | USA | 38 | i. Anxiety (75%)<br>ii. Depression (44%)<br>iii. Social Isolation (36%) | i. Occupational stress<br>ii. concerns for health and safety<br>iii. Additional work<br>iv. Psychological toll of caring for patients | I. Support and guide<br>ii. Policy changes |
| 120. | Nikolaidis et al., 2021 [147] | USA, UK | 3423 | i. Worry US (8–10%), UK (12–17%)<br>ii. Mood changes (50–57%) | With age and sex, Mood States | Government support |
| 121. | Rahman et al., 2021 [148] | 17 Asian countries | 8559 | i. Psychological distress (69%)<br>ii. Fear (24%) | i. Old age<br>ii. Poor financial status<br>iii. Nurses | Medical and social support |
| 122. | van Mulukom et al., 2021 [149] | 79 Countries | 8229 | i. Anxiety (8%)<br>ii. Depression (7%) | i. Self-isolation<br>ii. Poor coping strategy | i. Positive coping strategy<br>ii. Government support |

**Table 2.** Summary of Mental Disorders and Number of Studies Reported.

| S/N | Mental Disorder | No. of Studies that Reported | Percentage |
|---|---|---|---|
| 1 | Anxiety | 92 | 35.9 |
| 2 | Depression | 76 | 29.7 |
| 3 | Stress | 32 | 12.5 |
| 4 | Posttraumatic Stress Disorder | 16 | 6.3 |
| 5 | Psychological Disturbance | 14 | 5.5 |
| 6 | Insomnia | 13 | 5.0 |
| 7 | Worry | 6 | 2.3 |
| 8 | Fear | 5 | 2.0 |
| 9 | Obsessive-Compulsive Disorder | 1 | 0.4 |
| 10 | Eating Disorder | 1 | 0.4 |
|  | Total | 256 | 100% |

## 4. Results

In this review, a number of surveys outcome were summarized, including the prevalence, risks, and protective factors for mental health disorders (Table 1). A survey conducted in China reported an increased incidence of anxiety (45%), depression (50%), and insomnia (34%) [32]. Another longitudinal study indicated an increased level of anxiety (53%), depression (56%), and insomnia (79%) [33]. A study conducted in Australia reported a high rate of anxiety (40%), psychological distress (48%), and insomnia (41%) [34]. An online survey conducted in Malaysia also revealed a high level of anxiety (55%), depression (59%), and stress (31%) [35]. A study conducted in Iran indicated an increased incidence of anxiety (43%), depression (45%), and stress 35% [36]. Another study from Canada accounted for high level of anxiety (52%) [3]. Nonetheless, a quantitative online survey conducted in Germany reported a low incidence of anxiety (11%), depression (24%), and PTSD 5%) [37]. Another longitudinal survey from Spain also accounted for a moderate prevalence of anxiety (21%), depression (34%), and stress (28%) [38]. This is in line with other reviews that reported a global survey conducted among 31 nations [11,148].

During this study, various articles reviewed reported a wide range of mental disorders associated with the COVID-19 pandemic. Anxiety was the most commonly reported disorder by 35.9% of the articles reviewed. This indicated that anxiety was the most frequently encountered mental disorder during the COVID-19 pandemic. Depression was reported by 29.7% of the studies reviewed, making it the second most documented mental disorder. Stress is another mental disorder moderately revealed by 12.5% of the publications reviewed. However, PSTD, psychological disturbance, and insomnia were reported by only 6.3%, 5.5%, and 5.0% of the articles published, respectively. Generally, low reports of 2.3% documented worry, and only 2% of the articles reviewed reported the incidence of fear. Lastly, both OCD and eating disorders were reported by only 0.4% of the articles reviewed, respectively. This made them rare mental disorders during the COVID-19 pandemic (Table 2). This result reflects recent findings from other studies [14–21].

## 5. Discussion

The public health measures taken during the COVID-19 pandemic to curb the spread of SARS-CoV-2 infection such as quarantine, self-isolation, and total lockdown have caused detrimental effects on mental health. This review focused on the states mostly affected by the SARS-CoV-2 infection across the globe. The novelty in this research is to find the prevalence of mental disorders and to see if the prevalence is related to lockdown or high incidence of infection. Countries strongly affected by the COVID-19 pandemic include the USA, India, Brazil, France, the UK, Russia, Turkey, Italy, Spain, Germany, Argentina, and Iran. These countries recorded more COVID-19 cases and fatalities than others [150]. Several countries such as China, Australia, Malaysia, Iran, Germany, Spain, and Canada adopted total lockdown as a countermeasure which strongly affected their mental well-being. Generally, this review reported a high prevalence of wide ranges of mental disorders.

These include anxiety, depression, stress, PSTD, psychological disturbance, insomnia, worry, fear, OCD, and eating disorders. Additionally, a number of negative consequences of total lockdowns, such as suicidal tendencies, alcohol and substance use, and stigmatization were documented. This has made mental disorders one of the areas of global public health concern. Thus, the outcome is in line with a recent meta-analysis comprising 146,139 subjects worldwide which established a high prevalence of anxiety, depression, insomnia, and PTSD [15]. In addition, a recent report by the Organization for the Economic Cooperation and Development (OECD) that mapped out the strategy to improve public health also reported a high prevalence of anxiety and depression ranging between 5–50% and 3–36.8%, respectively [9]. Several studies carried out, however, have found a moderate prevalence of anxiety and depression [41,45,57–59]. Furthermore, a number of articles documented a very low prevalence of anxiety and depression [57,60,75,77,119,140,149]. Notably, separate studies carried out to describe the mental health of frontline workers reported a high rate of anxiety, depression, and PTSD [11,17,151,152]. The causes identified include shortage of PPE, inadequate testing kits, increased hospital duration, increased workload, lack of social and moral support, fear of infection, and stigmatization. Overall, mental health disorders have led to poor work performance, absconding from duty, job loss, and even suicide [6,9,11,13,15,17,152]. These deleterious outcomes will have a negative impact on people's quality of life, and by extension the global economy. Consequently, there is a need for a renewed effort by the United Nations, WHO, and individual nations to address the menace of mental disorders by increasing public welfare. Notably, addressing the high prevalence of mental disorders will improve individuals' quality of life, reduce the burden of public health and prevent or minimize the incidence of suicide, alcohol and substance use.

The prevalence of stress varies from high to moderate to low incidences. In this review, an extremely high prevalence of stress was reported [32,47,88]. Also, a number of longitudinal surveys revealed a moderately high incidence of stress [36,72,80,90,94–96]. However, the outcome of various online studies on mental health illnesses reported a moderate prevalence of stress, while several other studies accounted for a very low prevalence of stress [54,57,58,64,67,80,84,92,119]. Although stress is widely reported, a higher percentage might have overlapped as one of the general symptoms of other mental disorders. On this note, governments and other relevant stakeholders need to make a quick move to address increased hardships through the distribution of palliatives and giving incentives to people that have lost their jobs.

The media in general, have played vital role in spreading information and misinformation about the COVID-19 pandemic. Notable memories of various scenes of bloody pneumonia witnessed in Wuhan, China; mass graves repeatedly broadcast in Italy, Brazil, and Argentina; and the cremation of dead bodies in India resulted in horrific thoughts and flashbacks across the globe known as PTSD. In line with this, several longitudinal surveys reported moderate incidences of PTSD [42,96,97,99,111,138,139]. Nonetheless, a number of articles reviewed reported only a low incidence of PTSD [33,37,49,82,120,140,143]. Insomnia is one of the most acute symptoms of all mental disorders. Consequently, a report of an extremely high incidence of of insomnia was documented [65]. In addition, several others surveys revealed moderate incidences of insomnia [11,13,34,43,59,61,62,116]. Whereas few other longitudinal studies found a low prevalence of insomnia [75,84,96,108]. The high prevalence of insomnia has led to the misuse of drugs and substance abuse in an attempt to induce sleep. There was also hype about the efficacy of some medicines, such as chloroquine and hydroxychloroquine [2]. In general, there was a dire need to strengthen the guidelines and regulations guiding drug prescriptions and dispensing to counter panic buying by the public. The government also needs to regulate the contents broadcast by the media houses and impose sanctions where necessary to curb the spread of unnecessary panic and fake news.

Confining everyone indoors to enforce lockdown during the early days of COVID-19 has resulted in indiscriminate alcohol and other substance use to alleviate boredom. Con-

sequently, in this review, a survey reported a high incidence of alcohol and substance use [121]. In addition, a number of surveys have found moderate incidences of substance use [138,139]. COVID-19 pandemic is a global phenomenon and has affected almost every country, which made it known to almost every community across the globe. Despite this, a high incidence of stigmatization was reported [36]. Although only a few studies investigated the rate of suicide thought as a direct consequence of total lockdown and loss of freedom, most of the articles reviewed reported only low prevalence of suicide [3,59,114,134,138,139]. Generally, there is a need to further investigate the incidences of suicide, possibly due to agony, hopelessness, and despair caused by grief, bereavement and domestic violence linked to the COVID-19 pandemic.

In the course of this review, several risk factors for developing mental illness were identified. These include fear of infection, history of mental illness, poor financial status, female gender, alcohol drinking, younger age, lack of experience, comorbidities, and physical disability [3,6,11,12,32,33,35–37,80,125]. Accordingly, the OECD advocated for the need to identify and alleviate these risk factors [9,153,154]. In addition, the risk factors should be studied extensively and included in the strategic plan in preparation for the future pandemic.

This study also identified several protective factors against developing mental disorders, including higher income, public awareness, psychological counseling, social support, and government support [3,6,11,12,32,33,35–37,80,125]. There is strong advocacy to support vulnerable groups such as young adults, females, and elderly people regarding education, employment, and mental health support [9,153,154]. Governments and stakeholders must prioritize these protective factors when providing emergency relief and intervention during COVID-19 and future pandemics. Several coping strategies were identified and assessed for their effectiveness in reducing the global burden of mental health during COVID-19. These include awareness of the disease, indoor physical activities, online games, music concerts, online classes, and lectures [1,6,9,15,153,154].

Furthermore, various government interventions tried during the previous pandemics were identified and evaluated to see if they could be repurposed. In addition, various preventive measures aimed at reducing the negative impacts of horrible news of COVID-19 disease via traditional and social media were documented. This will guide the general public and media houses in identifying correct sources of information to avoid fake news. Also, it will reduce the broadcast of horrible hospital scenes and terrifying burial grounds. Overall, these will help significantly to prevent and reduce mental disorders caused by the COVID-19 pandemic [1,6,9,15,153,154].

## 6. Conclusions

COVID-19 pandemic, has revolutionized the global approach to healthcare and social issues. Lockdown and movement restrictions imposed by various governments have significantly affected the mental well-being of individuals. During the COVID-19 pandemic, anxiety was the most prevalent mental disorder, followed by depression, stress, and insomnia. In addition, a moderate incidence of psychological disturbance and PTSD was documented. Alcohol and substance use, domestic violence, stigmatization, and suicidal tendencies have all been identified as direct consequences of lockdown. These problems have significantly affected individuals' well-being and shut down the global economy. The eminent risk factors for mental health disorders identified include fear of infection, history of mental illness, poor financial status, female gender, and alcohol drinking. The documented protective factors were the higher-income level, public awareness, psychological counseling, social and government support. The primary focus of the government and other policymakers should utilize these protective factors in providing palliatives and incentives to cushion the economic impact of the lockdown. Overall, there is a need to monitor the long-term impact of COVID-19 pandemic on mental health during and after the COVID-19 pandemic.

### 7. Limitations of this Research

i. Several studies focused on specific people like healthcare workers and parents with children aged 18 years instead of the general population. ii. The selected articles' methodology includes self-report questionnaires or an instrument with cut-off points for stress, anxiety, or depression scores. Hence not homogenous. iii. The articles did not assess the reported mental disorders using a definitive psychiatric diagnosis by a specialist.

### 8. Article Highlights

I. The review highlighted the origin of the COVID-19 pandemic and the virulent nature of the SARS-CoV-2 virus. II. The COVID-19 pandemic has ravaged the global and individual economies, skyrocketing the global poverty index. III. The high prevalence of anxiety, depression, insomnia, and PTSD were reported during the COVID-19 pandemic, which calls for the urgent need for action. IV. Major risk factors for developing mental disorders include fear of infection, history of mental illness, female gender, younger age, comorbidities, and physical disability. V. Mental health interventions during the COVID-19 pandemic should focus on effective risk communication, continuous testing and assessment, unique approaches for vulnerable groups, community partnerships, online health applications, government policies, public welfare, and funding.

**Author Contributions:** A.R.A. and M.A.T. designed the study, searched for all the relevant articles, printed them, and screened the article for their eligibility based on inclusion and exclusion criteria. J.O., I.H.S., A.B.R., M.A.R., S.Y.N., A.A., A.D. and F.A. reviewed the articles, and constructed Tables 1 and 2 and wrote the discussion section. A.A., M.I., S.K. and A.E. reviewed and wrote the literature and all the references. A.R.A., M.A.T., S.R., S.S. and M.H. developed the research idea, wrote an abstract, and edited the manuscript. All authors have read and agreed to the published version of the manuscript.

**Funding:** This research received no external funding.

**Institutional Review Board Statement:** No applicated.

**Informed Consent Statement:** No applicated.

**Data Availability Statement:** Data is contained within the article.

**Acknowledgments:** We are grateful to the Department of Pharmacology and Therapeutics, Faculty of Pharmaceutical Sciences, Bayero University, Kano, Nigeria.

**Conflicts of Interest:** The authors declared no conflict of interest.

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
