# Peer review of "Challenges and Implications of the COVID-19 Pandemic on Mental Health: A Systematic Review"

_psych, doi:10.3390/psych4030035_

Round 1

Reviewer 1 Report

Dear Author,

Thank you for the opportunity to reflect on the manuscript. The topic of the manuscript is of importance and relevant methodology was used to investigate it. However, in the current version, the manuscript contribution to the readers is limited. The introduction section should be reorganized to avoid repetition within the section and in the findings. The methods section provides a partial explanation of the procedures. The results section is incomplete and contains in fact 2 tables all inclusive.  No meaningful discussion was done on the findings. The conclusions of the study can be hardly based on the findings. Please see my additional comments below. I hope the suggestions will be useful to enhance the presentation of your study.

Abstract

Lines 94-95: “These findings strongly …” - The association between the conclusion and the study findings is not clear. Please rephrase the last sentence to be a logical continuum of your findings.

Introduction

In general, there are many repetitions in the introduction and it looks like the results of the study. Please provide a background for the study need and importance and avoid dealing with the actual findings.  

Lines 107-108: “Scientists have yet to establish whether SARS-CoV-2 infection in the brain and not lockdown could cause neurodegenerative or mental diseases”. Please rephrase the sentence. It is very difficult to read it.

Line 108: “However…” this does not seem to be an appropriate connection between the two sentences. It sounds like you continue with the same line.

Lines 109-112: “Furthermore…” Again, the relation between the two sentences is not clear. You provide actually the information on longitudinal neurological impact of COVID-19. If so, please clarify what is the novelty of your study.

Lines 126:” Several reviews and meta-analyses..” The difference between this sentence and the previous ones (“Several people developed …”) is not clear. Please re-write.

Lines 128-140: You provide detailed information on the issue, that was previously mentioned in the first sentence. Please re-organize the introduction to avoid repetitions.

Methods

Line 169: Did you mean Figure 2?

Line 164-165: “The quality of the article retrieved was not examined using Newcastle Ottawa Scale..” But which scale/tools did you use to assure the studied quality? In addition, it is not clear why the quality of preprinted manuscripts can’t be evaluated?

You reported in Line 184 on the studies exclusion based on their methodology. Please detail the methodological issues which led to exclusion.

Results

It is not clear what the results are. The table 1 provides structured information, however, its conceptualization is not clear and its summary is missing. In fact, you should add the results section over and above the tables.

Table 2: The calculation of the percent is not clear. In addition, it is not clear what is the added value of this table over the Table 1.

Discussion

Most of the Discussion section is a repeat of the findings. No substantive discussion was made on either the findings or the implications of the consequences of the findings.

Author Response

REVIEWER I  

Abstract

  1. The sentence in Lines 94-95: “These findings strongly …” was paraphrased.

Introduction

  1. The repetitions in the introduction were removed.
  2. The sentence in Lines 107-108 was paraphrased and made readable.
  3. “However…” in Line 108 was replaced with ‘Therefore’ to make the sentence a continuous explanation.
  4. The whole sentence in Lines 109-112 with “Furthermore…” was removed.
  5. The repetition of ‘several’ in Lines 126 was removed and the sentence paraphrased.
  6. Information in Lines 128-140 on the neurological effect of COVID-19 was detailed to show relationship between COVID-19 and mental disorders.

Methods

  1. The figure mentioned in Line 169 was repositioned to Figure 3.
  2. The quality of the articles retrieved as mentioned in Line 164-165 were examined using Newcastle Ottawa Scale except for the pre-printed articles. The method used for examining the quality of the pre-printed articles was explained in details.
  3. The quality of the pre-printed articles was evaluated and the method used explained.
  4. The methodological issue that lead to exclusion of some articles as mentioned in Line 184 were listed.

Results

  1. Results in sentences were added above the table 1.
  2. Table 1 gives the exact prevalence of the mental disorders reported by each article.
  3. Table 2 summarized table 1 by giving the number of articles that reported each mental disorders (The frequency).
  4. Calculation in the Table 2 was corrected
  5. Table 2 showed anxiety was the most frequently reported mental disorder followed by depression etc… as added value.

Discussion

  1. Repeat of the findings was moved to the result section
  2. Substantive discussion was made
  3. Findings and the implications for the future were stated

REVIEWER II

  1. The authors retrieved a number of articles and after making comparism some were the same and were removed as duplicates as shown in figure 2.
  2. The articles that were searched manually from the references of the articles retrieved initially were now mentioned in the methodology.
  3. The manual search procedure was explained in the methodology.
  4. The reference sections of all the articles searched were manually checked and included afterwards.
  5. A total of 39 articles were obtained through manual search and now indicated in the methodology.
  6. Homogeneous instruments (anxiety and depression scale with a definite scores) were used for the assessment of anxiety and depression.
  7. Self-report instruments were used to determine the prevalence of the rest of mental disorders.
  8. No presence of a psychiatric diagnosis was considered in assessment of the reported mental disorders.
  9. The instruments used for anxiety, depression and stress have cut-off points.
  10. Heterogeneity in the method used to measure the presence of these disorders was mentioned as the limitations of the study.
  11. The title of Figure 3 which is now changed to figure 2 was edited as mental disorders and impact of lockdown associated with COVID-19.
  12. Although no article reported mental illness specifically due to having experienced the loss of loved ones or not having been able to carry out funeral rituals, however, they mentioned having infected family member, contact with a confirmed case or treating confirmed case which may eventually die as risk factors.
  13. All other mental disorders were used as search term and now included.
  14. Authors would consider making comparisons between countries with different measures against COVID-19 as future lines of work.

AUTHORS’ CONTRIBUTION

Abdullahi Rabiu Abubakar and Maryam Abba Tor designed the study, searched for all the relevant articles, printed them and screened the article for their eligibility based on inclusion and exclusion criteria. Joyce Ogidigo, Ibrahim Haruna Sani, Adekunle Babajide Rowaiye, Ramalan Aliyu Mansur, Sani Yahaya Najib, Ahmed Danbala and Fatima Adamu reviewed the articles, and constructed table 1 and 2 and wrote the discussion section. Adnan Abdullah, Mohammed Irfan, Santosh Kumar, and Ayukafangha Etando review and wrote the literature and all the references. Abdullahi Rabiu Abubakar and Maryam Abba Tor, Sayeeda Rahman, Susmita Sinha, and Mainul Haque developed the research idea, wrote an abstract, and edited the manuscript. All the authors read and approved the final manuscript for publication.

Reviewer 2 Report

I am grateful for the opportunity to review this interesting article.

The objective was to review published articles on mental disorders due to the COVID-19 pandemic, and associated risk and protective factors.

The paper has a clear objective and the methodology is adequate and the broad representation of countries in the selected articles is noteworthy. The discussion is well presented and the conclusions are relevant, highlighting the relevance of further research, on the medium and long-term effects and on the strategies to be adopted.

However, I would like to raise some questions about the method and make some observations on the results and the discussion.

Regarding the method, I would like the authors to provide more information on two aspects:

- In figure 2, there is no correspondence between the articles identified in the initial search by the two main authors (n=139 and n=106). If both authors used the same search criteria, how do they arrive at different results, and were the manually selected articles included here?

- The authors state that "manual search of reference sections of the suitable papers was conducted to identify studies not found through the database searches". Could the authors elaborate on how this manual search was done? Did they search all articles? What criteria did they use to consider which articles should be consulted and which should not? How many articles were included through this method?

The authors use the term "mental disorder".  In this regard, I would like to make two comments:

- How was the presence of anxiety, stress or depression assessed? were they disorders diagnosed by mental health specialists? were homogeneous instruments used for the assessment? were they self-report instruments? was the presence of a psychiatric diagnosis considered or were manifestations or symptomatology simply measured? were instruments with cut-off points and adapted to the sample used? was there homogeneity in the method used to measure the presence of these disorders? If the way of identifying the presence of disorders or symptomatology is not homogeneous, this should be clearly reflected in the limitations.

- In this sense, for example, Figure 3 considers Mental disorders assiciated qwith COVID-19. These include anger, worry, boredem or mode changes. While these may be considered mental health problems or factors that interfere with a person's well-being, they do not necessarily represent mental disorders.

On the other hand, it is striking that none of the studies include as risk factors variables such as having experienced the loss of one or more significant people due to COVID-19, or not having been able to carry out funeral rituals or accompany the deceased in their end-of-life process due to problems of confinement. It is also striking that none of the studies have identified an increase in the number of complicated bereavements as an effect of the pandemic. I would like the authors to explain whether it is possible that the inclusion of "anxiety", "depression", "stress" and "post-traumatic stress" as search terms may have made it easier for the publications identified to include precisely these disorders, and less likely others.

Finally, the authors could consider making comparisons between countries with different measures against COVID-19, especially confinement measures, or according to the phase of the pandemic in which the data are collected, as future lines of work.

In summary, this is a study with an interesting objective and an appropriate methodology, although it would be useful if the authors could provide more information on the aspects raised above.

Author Response

(The authors gave the same response as above.)

Round 2

Reviewer 1 Report

I would like to thank the authors for their detailed response on my comments. The manuscript was significantly improved from the previous version. However, some further changes are needed.

1.       The introduction section should include the rational for the study. Why it is important to delineate the mental health issues related to the COVID-19? Due to high burden of mental illness or/and there are other reasons? The longitudinal follow- up for COVID-19 mental health consequences is not really available now. Did you have some information from previous epidemics on mental health issues that can support your study rational? Are mental health issues conditional in the situation of epidemic or do they have a long term impact?

2.       Discussion:

-          Please add a clear statement on what is a novelty and contribution of your findings.

-          The conclusion of your manuscript addresses the need to “monitor the long-term impact of lockdown on mental health during and after the COVID-19 pandemic”. Please add literature through the discussion providing support for your conclusion. Why high level of stress, depression, PTSD, etc. rises an urgent call for action of governments and health authorities?

Author Response

  1. The introduction
  2. Rationale for the study was included.
  3. Mental health issues related to the COVID-19 is important because the preventive measures such as lockdown, hand washing, facemask, and closure of public places can lead to stress, alcohol and substance use and eventually mental illness.

iii. Information from previous epidemics on mental and neurological impact of viral infection were indicated

  1. Mental health issues usual occur acutely but the ability of the viral to penetrate and infect brain may result to the long term impact.
  2. Discussion:
  3. A clear statement on what is a novelty and contribution was added.
  4. The conclusion was paraphrased and changed from “monitor the long-term impact of lockdown on mental health” to “monitor the long-term impact of COVID-19 pandemic on mental health”

iii. Government and health authority’s intervention will reduce the prevalence of mental disorders and prevent suicide and related adversaries.
